# Evaluation of a ddPCR Commercial Assay for the Absolute Quantification of the Monkeypox Virus West Africa in Clinical Samples

**DOI:** 10.3390/diagnostics13071349

**Published:** 2023-04-04

**Authors:** Elena Pomari, Antonio Mori, Silvia Accordini, Annalisa Donini, Maddalena Cordioli, Evelina Tacconelli, Concetta Castilletti

**Affiliations:** 1Department of Infectious, Tropical Diseases and Microbiology, IRCCS Sacro Cuore Don Calabria Hospital, Negrar di Valpolicella, 37024 Verona, Italy; antonio.mori@sacrocuore.it (A.M.); silvia.accordini@sacrocuore.it (S.A.); annalisa.donini@sacrocuore.it (A.D.); concetta.castilletti@sacrocuore.it (C.C.); 2Division of Infectious Diseases, Department of Diagnostic and Public Health, University of Verona, 37100 Verona, Italy; maddalena.cordioli@aovr.veneto.it; 3Division of Infectious Diseases, Department of Medicine, Verona University Hospital, 37100 Verona, Italy; evelina.tacconelli@univr.it

**Keywords:** ddPCR, quantification, DNA, monkeypox virus

## Abstract

Background: Monkeypox virus (MPXV) is a double-stranded DNA virus belonging to the orthopoxvirus genus in the family *Poxviridae*. Distinct clades are identified: the clade I belonging to the Central African (or Congo Basin) clade and the subclades IIa and IIb belonging to the West African clade. Here, a commercial droplet digital PCR (ddPCR) assay was evaluated for the quantification of the MPXV West Africa clade in clinical samples. Methods: The ddPCR reaction was assessed as a duplex assay using *RPP30* as an internal amplification control. A total of 60 clinical specimens were tested, 40 positives (skin lesions, *n*=10; rectal swabs, *n* = 10; pharyngeal swabs, *n* = 10; and whole blood, *n* = 10), and 20 negatives (*n* = 5 for each biological matrix) were found at the routine molecular diagnostics (orthopoxvirus qPCR followed by confirmation with Sanger sequencing). To evaluate the analytical sensitivity, the ddPCR reaction was first analyzed on serial dilutions of synthetic DNA spiked in water and in negative biological matrices, achieving a limit of detection of 3.5 copy/µL. Results: Regarding the clinical samples, compared to routine molecular diagnostics, the ddPCR duplex assay showed 100% of specificity for all biological matrices and 100% sensitivity (10/10) for lesions, 100% (10/10) for rectal swabs, 90% (9/10) for pharyngeal swabs, and 60% (6/10) for whole blood. Conclusion: Overall, our data showed that the commercial ddPCR assay allowed the DNA detection of MPXV in 87.5% (35/40) of our cohort, highlighting useful technical indications for the different specimens with a potential greatest performance for skin lesions and rectal swabs.

## 1. Introduction

Monkeypox virus (MPXV) is an orthopoxvirus that causes a disease with symptoms similar, but less severe, to smallpox [1,2]. MPVX still occurs in countries in central and west Africa. Distinct clades are identified: the clade I belonging to the Central African (or Congo Basin) clade and the subclades IIa and IIb belonging to the West African clade [3,4,5]. MPVX is a zoonosis that can be acquired through contact with bodily fluids, lesions on the skin or on internal mucosal surfaces, such as in the mouth or throat, respiratory droplets from infected animals or humans, or contaminated objects [6,7]. Although human-to-human transmission is traditionally limited, recent evidence indicates that MPXV has found a novel route to transmit between humans [8,9,10,11]. In May 2022, there was the identification of clusters of MPVX cases in several non-endemic countries, leading to the “WHO Declaration of Monkeypox” as a global public health emergency [1,12]. Surveillance has expanded, and accurate case identification requires efficient routine diagnostic testing in order to control the spread of infection for its containment [13]. The most recent SARS-CoV-2 pandemic proved the significance of rapid testing, timely isolation, and contact tracing. Various commercial and in-house PCR-based diagnostic test assays have been developed in order to enhance the diagnostic capacity as well as test different specimen types [6,13,14]. MPXV is a double-stranded DNA virus, and detection of viral DNA by polymerase chain reaction (PCR) is the preferred laboratory test due to its higher sensitivity and specificity. The best diagnostic specimens are obtained directly from the rash–skin, fluid, or crusts, or a biopsy where feasible. In addition, MPXV detection has been reported in the anus, throat, and blood [15,16]. Compared to the more commonly used qPCR, ddPCR provides absolute quantification of DNA copies in samples without the need for standard curves and has the ability to detect lower genomic yields. DdPCR is based on microfluidics technology, which allows the generation of multiple reaction partitions that work as individual reactions. Based on the positive or negative fraction and following the Poisson distribution, it is possible to determine the absolute concentration of the target of interest in terms of the number of copies per microliter in the ddPCR reaction [17,18]. In this study, the commercial ddPCR assay (dEXD51818561, Bio-Rad) was evaluated for quantifying genomic DNA of the MPXV West Africa clade using four human biological matrices: skin lesions, rectal swabs, pharyngeal swabs, and whole blood. The performance characteristics evaluated were limit of detection (LoD), specificity, and sensitivity, which contributed to the support of the monitoring programs of MPXV in clinical samples.

## 2. Materials and Methods

### 2.1. Setting of the Study

A total of 60 clinical samples (skin lesion swabs, *n* = 15; rectal swabs, *n* = 15; pharyngeal swabs, *n* = 15; and EDTA whole blood, *n* = 15) collected at diagnosis or during follow-up of MPVX were used. Lesion (swabs of lesion surface and/or exudate), rectal, and pharyngeal specimens were collected using UTM or eSwab (COPAN). All the samples were previously screened by qualitative real-time PCR (qPCR) with routine diagnostic testing based on the RealStar orthopoxvirus PCR kit (RUO, Altona) (detecting orthopoxvirus species including MPXV) from May to August 2022. At diagnosis, Sanger sequencing for a fragment of the *crmB* gene [19] was performed to confirm MPXV infection (the confirmation was executed only in the lesion specimen or the sample with the lowest Ct value for each patient). Among the total number of samples, N = 40 were tested positive and N = 20 were tested negative for MPXV. All the positive cases were identified as West African clades with 100% identity on BLAST (https://blast.ncbi.nlm.nih.gov/Blast.cgi, accessed on 12 March 2023).

### 2.2. Automated DNA Extraction

DNA was isolated from 200 µL of transport medium and biological matrix by the EZ1 Advanced XL System using the EZ1 DSP Virus Kit (Qiagen), according to the manufacturer’s instructions. Samples were eluted in 90 μL. The isolated DNA was stored at −80 °C until use in ddPCR.

### 2.3. ddPCR

The ddPCR procedure was performed following the manufacturer’s instructions for the HEX probe assay for monkeypox in West Africa (dEXD51818561, not validated in clinical samples as specified by the manufacter Bio-Rad) targeting a fragment of 85nt of the *crmB* gene. In addition, the FAM probe assay for human *RPP30* (ddPCR CNV Assay Validated, 10031241, BioRad) was used as an internal amplification control (IAC). The analytical experiments were performed as simplex and duplex reactions. The simplex PCR reaction mixture was assembled as follows: 2x ddPCR Supermix for Probes (No dUTP) 11 μL, 20x Assay 2 μL, DNAse/RNase-free water 4μL, and DNA template 5 μL, for a final volume of 22 μL. The duplex PCR reaction mixture was assembled as follows: 2x ddPCR Supermix for Probes (No dUTP) 11 μL, 20x MPXV Assay 2 μL, 20x *RPP30* Assay 2 μL, DNAse/RNase-free water 2 μL, and DNA template 5 μL, for a final volume of 22 μL. Discordant samples were repeated using a 7 μL DNA template in duplicate (then merged for quantification). The QX200 droplet generator (Bio-Rad) was used to convert 20 μL of each reaction mix into droplets. The droplet-partitioned samples were transferred to a 96-well plate, sealed, and processed in a C1000 touch thermal cycler (Bio-Rad) under the following cycling protocol: 95 °C for 10 min for enzyme activation, 94 °C for 30 s for denaturation, and 55 °C for 60 s for annealing/extension for 40 cycles; 98 °C for 10 min for enzyme deactivation followed by an infinite 4-degree hold. The amplified samples were then transferred and read in the HEX (MPXV) and FAM (*RPP30*) channels using the QX200 reader (Bio-Rad). The experiments were performed using a negative control (no template control, NTC) and a positive control (a synthetic Gblock sequence following the manufacturer’s instructions). The reactions with less than 10,000 droplets and discordant results were repeated. The quantification was determined as at least 3 positive droplets for the reaction. Data were analyzed using the QXManager 1.2 Standard Edition Software (Bio-Rad) and expressed as copies/µL (cp/µL) in ddPCR reactions.

### 2.4. Limit of Detection Analysis

We used the synthetic amplicon Gblock Sequence (Appendix A) (following the manufacturer’s instructions) at 3.5 × 10^4^ copies/µL to generate 10-fold serial dilutions (from 10^−1^ to 10^−5^) for a total of 5 points in water, in the medium of transport (UTM and eSwab), and in negative (from healthy donors) lesions, rectal swabs, pharyngeal swabs, and whole blood. In particular, for lesion, rectal, and pharyngeal swabs, the analysis was conducted on two different types of swabs, UTM and eSwab (COPAN), that were used for our cohort of clinical samples. A sixth point was analyzed as a negative control (no DNA added) for water, the medium of transport, and each biological matrix. For the medium of transport and the biological matrices, the DNA was extracted with the automatic system described above. The duplex assay on biological matrices was performed in triplicate as independent experiments. Inter-assay CV% was calculated. For each analysis, results were expressed as cp/µL.

### 2.5. Statistics

Sensitivity and specificity for the clinical evaluation were analyzed by a two-way contingency table using a diagnostic test evaluation calculator (https://www.medcalc.org/calc/diagnostic_test.php, accessed on 20 February 2023).

## 3. Results

### 3.1. Limit of Detection Analysis and Assessment of Variability

In order to assess the LoD and variability of the commercial ddPCR MPXV assay used in duplex with *RPP30*, we first performed a 10-fold serial dilution of synthetic target DNA (3.5 × 10^4^ copies/µL) in water and in the medium of transport. In particular, we used two different types of media, UTM and eSwab. The viral copies were detected up to 3.5 cp/µL (dilution 10^−4^) in water and in both swab types (data are reported in Appendix A). No particular difference was observed in the quantification measurements for each point analyzed. No quantification was detected in the negative controls. Then, we performed the LoD analysis by ddPCR in spiked lesions, rectal swabs, pharyngeal swabs, and whole blood. We first conducted a simplex ddPCR as a preliminary test in order to verify MPXV measurements compared to the first analysis in water and transport media (described above) (data are reported in Appendix A). No particular difference was observed in the quantification for each point analyzed. Then, we assessed the duplex assay targeting MPXV and *RPP30* (Figure 1). The viral copies were detected up to 3.5 cp/µL (dilution 10^−4^) in all replicates for all samples (data are reported in Supplementary Material). In particular for MPXV, the linear regression coefficient R^2^ was >0.99 (Figure 2). Of note, the IAC *RPP30* was generally equal and with specific quantification dependently on the type of biological matrix and the used swab (UTM or eSwab). Basically, *RPP30* was detected at all points of dilution. The duplex assay for MPXV and *RPP30* was performed in triplicate. Concerning the variability of data, results were equally repeatable (data reported in Appendix A). Additionally, the coefficient of variation (CV) generally did not exceed 30% for all matrices and targets.

### 3.2. Clinical Samples

We investigated a total of 60 clinical samples, 40 positive (skin lesions, *n* = 10; rectal swabs, *n* = 10; pharyngeal swabs, *n* = 10; and whole bloods *n* = 10) and 20 negative (*n* = 5 for each biological matrix) at the routine qualitative orthopoxvirus (detecting of orthopoxvirus species including MPXV) qPCR followed by confirmation with Sanger sequencing (all the 40 positive cases were identified as West Africa clade with 100% of identity on BLAST). Table 1 summarizes the ddPCR results for each biological matrix. Supplementary material reports detailed data (Appendix A). Thus, in order to evaluate the clinical value, we calculated the sensitivity (SE) and specificity (SP) using the results obtained by our routine molecular diagnostic screening as reference values. Based on these data, the SE and SP of MPXV detection were respectively 100% [95% CI, 69.15 to 100.00] and 100% [95% CI, 47.82 to 100.00] for skin lesions (with the routine qPCR Ct value range of 18.41–33.01, Ct mean 25.57). For the rectal swab, the SE and SP were 100% [95% CI, 69.15 to 100.00] and 100% [95% CI, 47.82 to 100.00], respectively (with the routine qPCR Ct value range of 18.41–34.97, Ct mean 30.84). For the pharyngeal swab, the SE and SP were 90% [95% CI, 55.50 to 99.75] and 100% [95% CI, 47.82 to 100.00], respectively (with the routine qPCR Ct value range of 26.5–36.10, Ct mean 31.56). For whole blood, the SE and SP were 60% [95% CI, 26.24 to 87.84] and 100% [95% CI, 47.82 to 100.00] respectively (with the routine qPCR Ct value range of 32.61–39.79, Ct mean 35.14). Of note, we observed that the discordant results had a Ct value ≥ 34 (by the routine qPCR). In particular, only the quantification with ≥3 droplets was considered positive, and two of the whole blood were detected with <3 droplets. Some of the lesion specimens were repeated with a 1:10 dilution due to the high viral load. On the other hand, the majority of pharyngeal swabs and whole blood samples were repeated, increasing the DNA load. Concerning the IAC *RPP30*, it was detected in all samples of the total cohort. Of note, it highlighted a variable quantification in rectal and pharyngeal swabs. For all the experiments, negative and positive controls were used and verified.

## 4. Discussion

As of today, a few previous studies have reported the use of ddPCR for the detection of poxviruses [20,21,22]. Two recent works reported data obtained using *in-house* MPXV dPCRs for the quantification of in vitro cultured viruses [23] and of clinical samples [24]. In particular, Colavita et al. [24] reported the follow-up laboratory investigation of three MPXV cases infected in May-June 2022 from diagnosis to disease resolution, and the viral DNA widely varied in the different biological matrices investigated but was higher in oropharyngeal swabs, saliva, and stools compared to semen and urine. In order to support the monitoring programs of MPXV, the purpose of this study was to provide preliminary data about the evaluation (and not validation) of the commercial kit dEXD51818561 ddPCR West Africa assay (Bio-Rad) (validated in silico and not validated in clinical samples as specified by the manufacturer) on our available cohort of human samples (*n* = 60). These specimens were collected and analyzed for the diagnosis and follow-up of MPVX patients during the 2022 outbreak. In particular, we used four types of biological matrix: skin lesion swabs, rectal swabs, pharyngeal swabs, and whole blood. Lesion, rectal, and pharyngeal swabs were collected using two different types of items: eSwab and UTM. First, we compared the absolute quantification of synthetic MPXV DNA in the two media of transport with water, and no significant difference was observed in the measurements. Then, we compared the analysis of synthetic DNA using the different biological matrices. This analysis was performed as a simplex assay only for MPXV, as well as a duplex assay adding the IAC *RPP30*. The comparison showed agreement in MPXV measurements in both experiments.

Based on these data, we proceeded to analyze the duplex assay on the clinical samples. Overall the results showed 100% of specificity using the group of negative samples and 100% (10/10) of sensitivity for skin lesions, 100% (10/10) for rectal swabs, 90% (9/10) for pharyngeal swabs, and 60% (6/10) for whole blood in the group of the positives. It has been known that ddPCR has a low dynamic range, leading to the need for dilution for analysis [25]. This can be necessary for lesions, as the viral load is frequently higher. Indeed, some of our lesion specimens were diluted to avoid the saturation with signal. On the contrary, regarding the other specimens, and especially for the pharyngeal swab and whole blood, the increase in DNA load was necessary.

Of note, the RPP30 highlighted a variable quantification in lesion, rectal, and pharyngeal swabs, suggesting that the accuracy of the test can be negatively affected by the sample collection [26,27,28]. For the purpose of our study, we did not consider the use of additional reference genes for the different specimens analyzed. Indeed, we chose RPP30 as an indicator for cellular content [29,30] and in order to assess a simple duplex assay to be performed in a unique reaction. So, a further investigation can be useful in order to verify this hypothesis using other internal amplification controls in addition to RPP30.

To conclude, in this work, the dEXD51818561 commercial ddPCR assay detected MPXV in 87.5% (35/40) of the entire positive cohort. For the pharyngeal swab, data were not totally equal in terms of SE compared to the routine qPCR, and further investigation might elucidate if the specimen collection could affect the detection rate, as shown from the variable *RPP30* quantification. On the other hand, the results from the skin lesion and rectal swab were more accurate in terms of SE and SP in the total cohort of samples, suggesting that these specimens are preferred for the absolute quantification of the virus. Moreover, the results from whole blood suggest that this type of specimen could support the detection of MPXV but with a low level of virus [6,14]. Thus, a larger number of samples and data from other laboratories are needed to evaluate the clinical value of ddPCR. Further investigations will be necessary, focusing on the assessment of the performance of the MPXV absolute quantification (*i*) compared with quantitative qPCR and (*ii*) using additional viral targets.

## Figures and Tables

**Figure 1 diagnostics-13-01349-f001:**
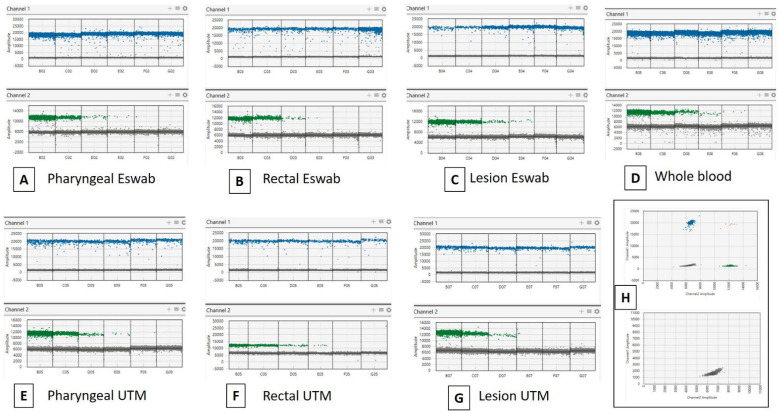
Evaluation of the ddPCR duplex assay (MPXV as Channel 1 and *RPP30* as Channel 2) performance using MPXV synthetic amplicon Gblock Sequence as reference DNA. Standard curves were obtained from 10-fold serial dilutions of the reference DNA (3.5 × 10^4^ copies/µL) in four biological matrices in two different transport media (UTM and eSwab) (**A**–**G**). Positive samples (up) and negative samples (below) (**H**).

**Figure 2 diagnostics-13-01349-f002:**
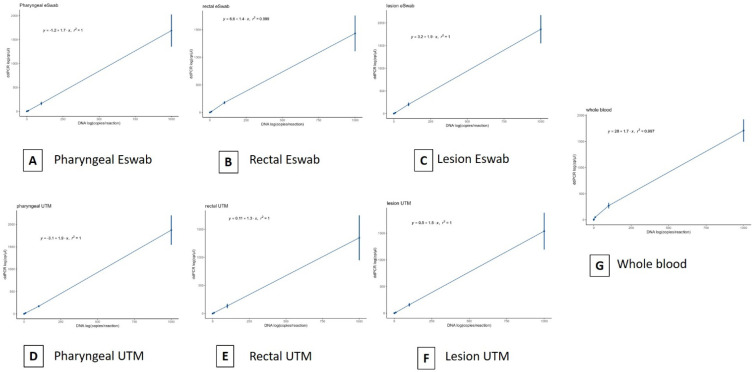
Evaluation of the ddPCR assay performance using MPXV synthetic amplicons and Gblock sequences as reference DNA. Standard curves were obtained from 10-fold serial dilutions of the reference DNA (3.5 × 10^4^ copies/µL) in four biological matrices in two different transport media (UTM and eSwab). The curves also include the negative point (no DNA). (**A**–**G**). The error bar represented the standard deviation of the mean. Pearson statistics were used.

**Table 1 diagnostics-13-01349-t001:** Summarized results in the clinical samples. CI—confidence interval; SE—sensitivity; SP—specificity.

		qPCR for Orthopoxvirus (Reference)			
Specimen	ddPCR for Crmb MPXV	Positive	Negative	SE [95% CI]	SP [95% CI]
Skin lesion	Positive	10	0	100% [69.15–100.00]	100% [47.82–100.00]
	Negative	0	5		
	Positive	10	0	100% [69.15–100.00]	100% [47.82–100.00]
Rectal swab	Negative	0	5		
	Positive	9	0		
Pharyngeal swab	Negative	1	5	90% [55.50–99.75]	100% [47.82–100.00]
	Positive	6	0	60% [26.24–87.84]	100% [47.82–100.00]
Whole blood	Negative	4	5		

## Data Availability

All data generated or analyzed during this study are included in this published article (and its Appendix A).

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
