# Peer review of "Evaluation of a ddPCR Commercial Assay for the Absolute Quantification of the Monkeypox Virus West Africa in Clinical Samples"

_diagnostics, 2023, doi:10.3390/diagnostics13071349_

Round 1

Reviewer 1 Report

1.It is not really convincing to employ one pair of primers to discuuss dPCR detection performance frorm different kinds of samples.  Especially, the primers haven't been tested in clinical samples.

2.It was reported that 2022 MPXV segregated in a divergent phylogenetic branch by genome analysis. I wonder wheather the dPCR methed in the paper could detect the latest MPXV.

3. When we talk about viral load, the development stage of disease should be considered. Have the authors noticed the clinical phases of patients whose samples were analyzed ?

4. I suggest that the authoros could analye the quantification resluts and norminal concentration of the diluted samples when talking about LODs of the method.

Author Response

Comments and Suggestions for Authors

1.It is not really convincing to employ one pair of primers to discuuss dPCR detection performance frorm different kinds of samples.  Especially, the primers haven't been tested in clinical samples.

We thank the Reviewer for the suggestion. The purpose of our work was to evaluate the dEXD51818561 commercial kit. For this reason, no other target genes have been considered. BioRad proposed us the assay as in silico validated and we decided to test our available cohort of clinical samples diagnosed for MPXV with routine lab diagnosis performed with real time PCR and Sanger sequencing (as specified in the manuscript). However, the need of using additional genes is highlighted in the conclusion of the manuscript.

2.It was reported that 2022 MPXV segregated in a divergent phylogenetic branch by genome analysis. I wonder wheather the dPCR methed in the paper could detect the latest MPXV.

We thank the Reviewer for the suggestion. As reported in the text of the manuscript, the commercial kit that we evaluated was in silico validated. The assay has been designed to detect a fragment of crmB, the target gene mostly used to detect Orthopoxviruses (https://doi.org/10.1128/JCM.39.1.94-100.2001). In particular, we tested our cohort before the American CDC and European ECDC lab alerts (referring to the C3L PCR assay by Li et al. 2010). Indeed, we analysed samples collected from May to August 2022 as specified in the text of the manuscript, so all belonging to IIb clade. For those biological matrices tested negative, the ddPCR assay detected MPXV in the other paired matrices (such as patients 1909 and 1998 -  suppl. material 2) suggesting that the assay is able to detect MPXV 2022 outbreak. Moreover, we aligned the amplicon sequence (kindly and confidentially provided by the manufacter but under intellectual property, that’s why we showed the entire Gblock in Figure S1) in BLAST and we found 100% match with sequences from recent MPXV collected in 2023. We now changed a phrase in the Discussion, original phrase “To conclude, in this work we evaluated that the commercial ddPCR assay allows the DNA quantification of MPXV”, new phrase: “To conclude, in this work the dEXD51818561 commercial ddPCR assay detected MPXV in 87.5% (35/40) of the entire positive cohort.” As well as we changed in the Abstract.

  1. When we talk about viral load, the development stage of disease should be considered. Have the authors noticed the clinical phases of patients whose samples were analyzed ?

In the present work, our purpose was not to analyse the viral load. We used an available cohort of samples collected from different patients, some at diagnosis and some during follow up (as specified in the text of the manuscript). We evaluated the commercial kit in the detection of MPXV in clinical samples independently of the clinical phase, as was performed in a previous article (cited in the manuscript) (Colavita F, Mazzotta V, Rozera G, Abbate I, Carletti F, Pinnetti C, Matusali G, Meschi S, Mondi A, Lapa D, Vita S, Minosse C, Aguglia C, Gagliardini R, Specchiarello E, Bettini A, Nicastri E, Girardi E, Vaia F, Antinori A, Maggi F. Kinetics of viral DNA in body fluids and antibody response in patients with acute Monkeypox virus infection. iScience;26(3):106102, 2023).

  1. I suggest that the authoros could analye the quantification resluts and norminal concentration of the diluted samples when talking about LODs of the method.

We agree with the Reviewer and we now added in the text and in the suppl. materials the quantity of LoD. 

Reviewer 2 Report

The commercial ddPCR assay of the DNA quantification of MPXV was evaluated in this manuscript

From the results, the applicability of this kit is limited. Compared with conventional quantitative PCR methods, ddPCR can only detect samples with Ct <34. The reasons that affect the results may come from the kit, samples, or nucleic acid extraction.

For the pharyngeal swab and the whole blood, data of SE were not totally equal to the routine rtPCR. This kit is not suitable for MPXV detection in whole blood

Author Response

Comments and Suggestions for Authors

The commercial ddPCR assay of the DNA quantification of MPXV was evaluated in this manuscript. 

From the results, the applicability of this kit is limited. Compared with conventional quantitative PCR methods, ddPCR can only detect samples with Ct <34. The reasons that affect the results may come from the kit, samples, or nucleic acid extraction.

For the pharyngeal swab and the whole blood, data of SE were not totally equal to the routine rtPCR. This kit is not suitable for MPXV detection in whole blood

We thank the Reviewer for the comment. We understand that no particular suggestions were highlighted. We are aware that it is usually preferable to use a sensitive qualitative method instead of a quantitative assay. We think that the Discussion of the manuscript highlighted all these issues.

Reviewer 3 Report

The reviewed manuscript is dedicated to the design and validation of ddPCR-based assay detecting mpox. Here, authors validated a commercial ddPCR approach using synthetic DNA controls and clinical samples. The presented results are timely and interesting for scientists, specializing on the field of molecular diagnostics. However, several issues need to be addressed before the publication of the manuscript. Please, find point-by-point comments below:

Major issues:

1.      Authors are encouraged to provide more background information about digital PCR for mpox diagnosis, describing previously published approaches for mpox testing with their limitations and advantages. Also, authors are requested to compare the reported ddPCR-based assays with the method presented here in terms of sensitivity and specificity.

2.      Graphs depicting relation between Ct values and ddPCR results would increase the readability of the manuscript. also, authors are encouraged to provide the ratio between mpox DNA load and human genomic DNA for various types of clinical samples.

Minor issues:

1.      Page 2, line 71: “Real-Time PCR (rtPCR)” — the MIQE guide recommends to avoid rtPCR as an abbreviation for real-time PCR as it can be confused with reverse-transcriptase PCR. Please, consider the usage of another abbreviation like “qPCR”.

Author Response

Comments and Suggestions for Authors

The reviewed manuscript is dedicated to the design and validation of ddPCR-based assay detecting mpox. Here, authors validated a commercial ddPCR approach using synthetic DNA controls and clinical samples. The presented results are timely and interesting for scientists, specializing on the field of molecular diagnostics. However, several issues need to be addressed before the publication of the manuscript. Please, find point-by-point comments below:

Major issues:

  1. Authors are encouraged to provide more background information about digital PCR for mpox diagnosis, describing previously published approaches for mpox testing with their limitations and advantages. Also, authors are requested to compare the reported ddPCR-based assays with the method presented here in terms of sensitivity and specificity.
  2. Graphs depicting relation between Ct values and ddPCR results would increase the readability of the manuscript. also, authors are encouraged to provide the ratio between mpox DNA load and human genomic DNA for various types of clinical samples.

We thank the Reviewer for the comments. About the background, so far only two papers have been published about dPCR for mpox as we reported in the discussion. We added a phrase to highlight the results, as suggested. Regarding the comparison of ddPCR with other methods, our work was not a validation but a clinical evaluation of the ddPCR commercial kit. Our real time PCR is qualitative and not quantitative (as we now specified in the text of the manuscript). For this reason we did not compare the two methods. Moreover, the ddPCR assay is specific for MPXV crmB, whilst the real time PCR assay is designed for Orthopoxvirus species including MPXV (RealStar Orthopoxvirus PCR kit (RUO, Altona) (target region not specified by the manufacter). The need of doing the comparison with quantitative qPCR is highlighted in the conclusion of the manuscript.

Minor issues:

  1. Page 2, line 71: “Real-Time PCR (rtPCR)” — the MIQE guide recommends to avoid rtPCR as an abbreviation for real-time PCR as it can be confused with reverse-transcriptase PCR. Please, consider the usage of another abbreviation like “qPCR”.

We agree with the Reviewer. We changed as suggested in the text of the manuscript.

Round 2

Reviewer 1 Report

no more suggestion

Reviewer 2 Report

No comments